# IndexNet: Timestamp and Variable-Aware Modeling for Time Series Forecasting

## Abstract

Multivariate time series forecasting (MTSF) plays a vital role in a wide range of real-world applications, such as weather prediction and traffic flow forecasting. Although recent advances have significantly improved the modeling of temporal dynamics and inter-variable dependencies, most existing methods overlook index-related descriptive information, such as timestamps and variable indices, which carry rich contextual semantics. To unlock the potential of such information and take advantage of the lightweight and powerful periodic capture ability of MLP-based architectures, we propose **IndexNet**, an MLP-based framework augmented with an **Index Embedding (IE) module**. The IE module consists of two key components: **Timestamp Embedding (TE)** and **Channel Embedding (CE)**. Specifically, TE transforms timestamps into embedding vectors and injects them into the input sequence, thereby improving the model's ability to capture long-term complex periodic patterns. In parallel, CE assigns each variable a unique and trainable identity embedding based on its index, allowing the model to explicitly distinguish between heterogeneous variables and avoid homogenized predictions when input sequences seem close. Extensive experiments on 12 diverse real-world datasets demonstrate that IndexNet achieves comparable performance across mainstream baselines, validating the effectiveness of our temporally and variably aware design. Moreover, plug-and-play experiments and visualization analyses further reveal that IndexNet exhibits strong generality and interpretability, two aspects that remain underexplored in current MTSF research. *Implementation details and reproducible code are provided in the supplementary materials.*

## 1 Introduction

Multivariate time series forecasting (MTSF) aims to predict future values from historical observations and plays a pivotal role in numerous real-world applications, including financial investment (Sezer et al., 2020), weather forecasting (Karevan & Suykens, 2020), and traffic flow prediction (Shu et al., 2021; Miao et al., 2024b). In such scenarios, temporal and variable index information, such as timestamps and variable indexes, provides rich semantic cues. For instance, timestamps naturally provide periodic patterns (Yue et al., 2022; Dyreson & Snodgrass, 1993), while variable indexes indicate different dynamics across heterogeneous variables (Shao et al., 2022).

Although timestamps and variable indexes are crucial for interpreting multivariate time series, existing MTSF methods still struggle to exploit them effectively (Dai et al., 2024a; Liu et al., 2024b). This drawback not only diminishes predictive accuracy but also raises issues of reliability and interpretability, both of which are vital in domains such as investment and decision-making (Ismail et al., 2020; Oreshkin et al., 2019; Lim et al., 2021; Fortuin et al., 2018). Reviewing earlier studies on timestamp modeling, as shown in Fig. 1, early efforts integrated timestamps by aligning them with sequences along the channel dimension and fusing them through addition in the latent space (Wu et al., 2021; Zhou et al., 2021; Wu et al., 2023), yet the gains were often marginal. A retrospective perspective (Wang et al., 2024a) indicates that projecting timestamp information across multiple channels drives models into overly complex multivariate interactions to extract complete temporal cues (Han et al., 2023). Subsequently, inspired by the strong performance of models (Zeng et al., 2023; Nie et al., 2023) without timestamps, many approaches abandoned them altogether. Although a few recent works have revisited timestamp modeling, the improvements remain limited.

Moreover, with respect to variable indices, although they naturally correspond to variable identities, their role has been largely overlooked in existing studies (Hu et al., 2025; Liu et al., 2024a).

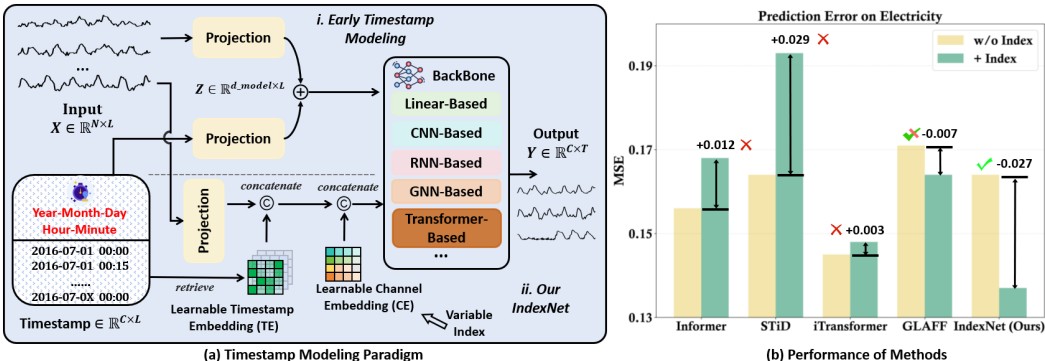

Figure 1: The general timestamp processing strategy in early works and its comparative experimental results with more recent methods. Subfigure (a) illustrates the typical processing of timestamp information before to modeling. Subfigure (b) compares the forecasting performance of different methods with timestamp on the Electricity dataset, with input and output sequence lengths are 96.

Incorporating identity information can substantially enhance models in characterizing heterogeneous variables, especially when they exhibit similar inputs but diverge in future dynamics. By assigning variable-specific identities, the model can effectively distinguish unique temporal patterns and produce tailored forecasts for each variable. For example, in weather forecasting, humidity and wind speed are fundamentally different variables with distinct temporal behaviors. Current approaches, however, often treat all variables as homogeneous sequences, either modeling them independently or relying on intricate mechanisms to capture inter-variable interactions and differences (Shao et al., 2024). The former strategy risks confusion, particularly when heterogeneous channels share similar inputs yet evolve differently, making it difficult for channel-independent models to account for variable-specific properties (Liang et al., 2022). The latter, in contrast, typically requires elaborate designs and complex network architectures, which tend to be fragile and computationally expensive.

To address these issues, we propose an index-aware network, termed **IndexNet**. IndexNet comprises two core components: an **Index Embedding (IE)** module and a lightweight MLP-based backbone. The IE module integrates index-related prior knowledge (e.g., timestamps and variable indices) through two submodules: **Timestamp Embedding (TE)** and **Channel Embedding (CE)**. The TE submodule injects temporal information into each sequence by constructing and retrieving a set of learnable timestamp embeddings, where each embedding encodes periodic patterns at a specific scale (e.g., day, hour, minute). Any time series can retrieve the corresponding embedding based on its timestamp, thereby improving predictive accuracy and reliability. The CE submodule captures variable-specific dynamics by encoding variable identities. It builds a group of learnable channel embeddings, assigning each variable a unique representation. Each variable retrieves the identity vector via its index for variable-aware modeling. Meanwhile, given the efficiency, generalization, and periodic modeling capacity of MLPs (Li et al., 2023), an MLP-based backbone is employed for representation learning. Through this design, **IndexNet** not only preserves the unique advantages of MLPs in sequence modeling but also effectively exploits both timestamp and variable identity information, delivering a lightweight, robust, and interpretable solution for MTSF.

In a nutshell, the contributions of our paper are summarized as follows:

- We propose an index-aware network, named **IndexNet**, which leverages both timestamp and variable identity information to provide a robust and interpretable solution for MTSF, addressing the long-standing neglect of index-related cues in existing models.

- The **Index Embedding (IE)** module contains two subcomponents: **Timestamp Embedding (TE)**, which injects periodic information into sequences to enhance reliability and interpretability; and **Channel Embedding (CE)**, which assigns each variable a distinct learnable identity vector, enabling the model to capture variable-specific patterns.

- Experiments on 12 public MTSF datasets show that IndexNet achieves competitive performance against recent methods. In addition, plug-and-play and visualization results highlight the generality and interpretability of our approach, offering meaningful temporal- and variable-wise insights.

## 2 RELATED WORK

### 2.1 MULTIVARIATE TIME SERIES FORECASTING

Multivariate time series forecasting (MTSF) has garnered significant attention due to its wide applicability in real-world scenarios (Dai et al., 2024b; Liu et al., 2024c; Nie et al., 2023). Classical statistical approaches, such as ARIMA (Nelson, 1998) model temporal dynamics via autoregressive and moving average components. However, the complexity and dynamics of real-world data often challenge the adaptability of such methods (Gough et al., 2010; Ramana et al., 2000). With the rapid progress of deep neural networks (DNNs) in domains such as natural language processing and computer vision, DNN-based models have emerged as a dominant paradigm in MTSF. These methods aim to capture intricate dependencies across multiple variables through carefully designed network architectures. Recent research broadly categorizes these models into **Channel Independent (CI)** and **Channel Dependent (CD)** approaches. CI methods (Zeng et al., 2023; Das et al., 2023; Nie et al., 2023; Dai et al., 2024a; Lin et al., 2024; Miao et al., 2025) make predictions solely based on the historical values of each individual variable, deliberately avoiding inter-variable interactions and timestamps. This design simplifies the learning process, stabilizes training, and excels at capturing rapid, channel-specific temporal dynamics. In contrast, CD methods (Wu et al., 2021; Zhou et al., 2022; Wu et al., 2023; Zhang & Yan, 2023; Liu et al., 2024c), predominantly Transformer-based, explicitly model the correlations among different variables to exploit cross-channel dependencies. While these models incorporate richer information, they often suffer from severe overfitting when capturing complex inter-variable relationships, leading to marginal improvements or even performance degradation in practical forecasting tasks.

### 2.2 TIMESTAMP AND VARIABLE INDEX

Early deep learning-based models for multivariate time series forecasting typically adopt complex encoder-decoder architectures with embedding layers. Representative models such as Informer (Zhou et al., 2021), TimesNet (Wu et al., 2023), and others (Zhou et al., 2022; Liu et al., 2021) incorporate timestamp information by adding timestamp embeddings to positional and value embeddings. While this approach helps extract temporal patterns from raw observations, it entangles timestamp semantics across all variable channels, forcing the model to implicitly learn timestamp-variable interactions. This often leads to overfitting and limited generalization (Wang et al., 2024a). Later studies, such as DLinear (Zeng et al., 2023), demonstrated that simple linear models without timestamp inputs or inter-variable modeling could outperform these complex frameworks, highlighting the strength of lightweight architectures and questioning the utility of timestamp encoding.

As a result, recent methods (Nie et al., 2023; Lin et al., 2024) have removed timestamp inputs altogether. However, this overlooks the semantic value of timestamps in varying order, periodicity, and seasonality—key elements in real-world temporal signals. Some recent approaches have revisited timestamp modeling from new angles. For example, (Shao et al., 2022) introduces temporal indices at intermediate layers, achieving gains in limited settings. iTransformer (Liu et al., 2024c) embeds timestamp features into attention tokens, though improvements remain marginal. Similarly, GLAFF (Wang et al., 2024a) integrates timestamp semantics at the decoder stage but fails to deliver consistent benefits across datasets. In contrast, variable index information remains largely underutilized in MTSF. Variable indexes encode variable-specific identities and can provide valuable priors for disentangling heterogeneous temporal dynamics. Despite this, only a few models Shao et al. (2022); Lin et al. (2023) explicitly incorporate variable index cues, leaving a gap in the development of identity-aware forecasting approaches.

## 3 METHOD

In multivariate time series forecasting, the goal is to predict the future sequence $\mathbf{Y} = [\mathbf{x}_{L+1}, \ldots, \mathbf{x}_{L+T}] \in \mathbb{R}^{N \times T}$ given the historical input sequence $\mathbf{X} = [\mathbf{x}_1, \ldots, \mathbf{x}_L] \in \mathbb{R}^{N \times L}$, where $L$ and $T$ denote the lengths of the input and output sequences respectively, and $N$ represents the number of variables. It is important to note that the timestamp information corresponding to each time step varies according to the sampling interval and temporal resolution of the dataset. For example, when data is recorded at an hourly resolution spanning multiple years, each timestamp generally consists of four components: year, month, day, and hour, which together form a timestamp sequence $\mathbf{TS} = [\mathbf{S}_1, \ldots, \mathbf{S}_L] \in \mathbb{R}^{4 \times L}$.

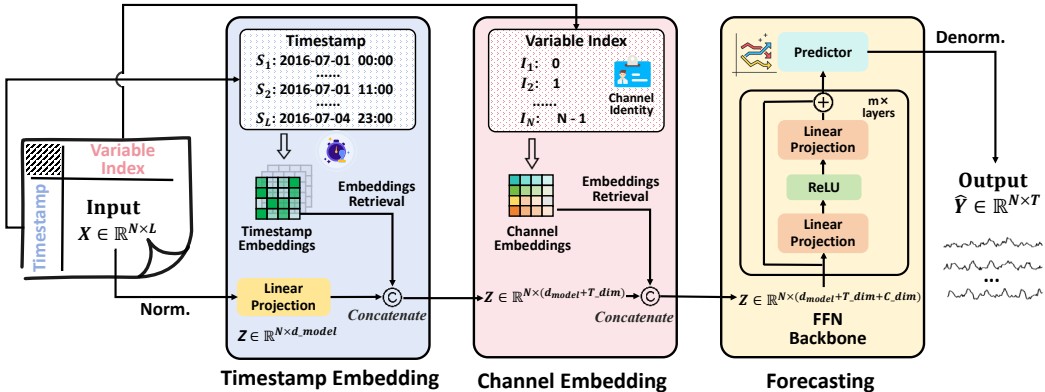

Figure 2: The overall architecture of IndexNet, which consists of three modules: Timestamp Embedding, Channel Embedding, and Forecasting.

### 3.1 STRUCTURE OVERVIEW

As illustrated in Fig. 2, IndexNet consists of three modules: **Timestamp Embedding (TE)**, **Channel Embedding (CE)**, and **Forecasting**. Given a multivariate input sequence, IndexNet first normalizes the raw data, which is then combined with index-related information throughout the pipeline. The TE module generates learnable embeddings for timestamp components such as day, hour, and minute, injecting temporal semantics into each time step by retrieving and integrating the corresponding embeddings based on the timestamp values. After that, the CE module assigns each variable a unique, learnable identity vector, which is concatenated with the input features following a linear projection, explicitly encoding variable-specific information. Finally, this enriched representation is fed into the Forecasting module—an encoder composed of multiple MLP layers—that models temporal dynamics and produces the forecast through a final linear projection and de-normalization. Each of these modules will be described in detail in the following sections.

### 3.2 DATA PREPROCESSING

We first apply Z-score normalization (Kim et al., 2022) to the input $\mathbf{X} = [\mathbf{x}_1, \ldots, \mathbf{x}_N] \in \mathbb{R}^{N \times L}$ to obtain normalized sequence $\hat{\mathbf{x}}_n$. Later, we construct a timestamp sequence $\mathbf{TS} = [\mathbf{S}_1, \ldots, \mathbf{S}_L] \in \mathbb{R}^{k \times L}$, where $k$ is the dimensionality of each timestamp. This sequence is subsequently fed into the Timestamp Embedding module. If explicit timestamps are unavailable in the original dataset, we use sequential indices along the time axis as a proxy. Assuming a sampling interval of one hour, we extract coarse-grained periodic time features such as the *hour of day* and *day of week*:

$$\text{HourOfDay}(t) = t \bmod 24, \quad \text{HourOfWeek}(t) = \left\lfloor \frac{t}{24} \right\rfloor \bmod 7,$$

$$\mathbf{S}_t = \text{Concatenate}(\text{HourOfDay}(t), \text{HourOfWeek}(t)),$$

where $t \in \{0, 1, \ldots, H - 1\}$, and $H$ is the number of time steps in the dataset; 24 and 7 correspond to the number of hours in a day and days in a week, respectively. The extracted $\text{HourOfDay}(t) \in \{0, 1, \ldots, 23\}$ and $\text{HourOfWeek}(t) \in \{0, 1, \ldots, 6\}$ features are concatenated to form the timestamp representation $\mathbf{S}_t$ for each time step $t$. This encoding assigns each time step a periodic index that captures cyclical temporal patterns as well as the relative position within the sequence.

### 3.3 TIMESTAMP EMBEDDING

To encode periodic temporal semantics, the **Timestamp Embedding (TE)** module constructs multiple sets of learnable embeddings, each corresponding to a specific temporal feature such as minute, hour, day of week, and so forth. Each set contains embedding vectors whose number matches the cardinality of the respective temporal feature (e.g., 7 vectors for *day of week*, 24 for *hour of day*). Each embedding vector has a dimensionality equal to the input sequence length $L$, ensuring proper alignment of embeddings with each time step. Formally, the embedding matrices are defined as:

$$\mathbf{E}^{\text{minute}} \in \mathbb{R}^{K \times T_{dim}}, \quad \mathbf{E}^{\text{hour}} \in \mathbb{R}^{24 \times T_{dim}}, \quad \mathbf{E}^{\text{dow}} \in \mathbb{R}^{7 \times T_{dim}},$$

$$\mathbf{E}^{\text{dom}} \in \mathbb{R}^{31 \times T_{dim}}, \quad \mathbf{E}^{\text{month}} \in \mathbb{R}^{12 \times T_{dim}},$$

where $T_{dim}$ denotes the latent dimension of the feature. $\mathbf{E}^{\text{minute}}, \mathbf{E}^{\text{hour}}$, and $\mathbf{E}^{\text{dow}}$ correspond to the minute of an hour, hour of a day, and day of a week, respectively. For finer granularity, $\mathbf{E}^{\text{minute}}$ encodes minute-level features, where $K = \frac{60}{\tau}$ specifies the number of minute intervals per hour given a sampling interval of $\tau$ minutes. In contrast, $\mathbf{E}^{\text{dom}}$ and $\mathbf{E}^{\text{month}}$ capture coarser month-level patterns. All embedding matrices are initialized to zero to promote stable training.

Based on the constructed groups of timestamp embeddings, we retrieve temporal representations by indexing into the corresponding timestamp embeddings using the discrete values from the timestamp sequence. Given a timestamp input sequence $\mathbf{TS} = [\mathbf{S}_1, \ldots, \mathbf{S}_L] \in \mathbb{R}^{d \times L}$, where each $\mathbf{S}_t$ contains $d$ discrete temporal fields (e.g., minute, hour, weekday), the corresponding embeddings are retrieved as follows:

$$\mathbf{e_t}^{\text{minute}} = \mathbf{E}^{\text{minute}}[\text{Minute}(t)], \quad \mathbf{e_t}^{\text{hour}} = \mathbf{E}^{\text{hour}}[\text{Hour}(t)], \quad \mathbf{e_t}^{\text{dow}} = \mathbf{E}^{\text{dow}}[\text{DayofWeek}(t)],$$

$$\mathbf{e_t}^{\text{dom}} = \mathbf{E}^{\text{dom}}[\text{DayofMonth}(t)], \quad \mathbf{e_t}^{\text{month}} = \mathbf{E}^{\text{month}}[\text{Month}(t)],$$

where the functions $\text{Minute}(t)$, $\text{Hour}(t)$, and $\text{DayofWeek}(t)$ extract the integer index values for the respective temporal features from the timestamp vector $\mathbf{S}_t$ at time step $t$. The retrieved embedding vectors $\mathbf{e_t}^{\text{minute}}, \mathbf{e_t}^{\text{hour}}, \mathbf{e_t}^{\text{dow}}, \mathbf{e_t}^{\text{dom}}, \mathbf{e_t}^{\text{month}}$ represent the timestamp embeddings aligned with the input sequence length. When the sampling granularity $\tau = 60$ minutes (i.e., hourly sampling), the minute-level embedding is omitted since it provides no additional temporal resolution. These embeddings are then aggregated to form the final timestamp representation as follows:

$$\mathbf{e}_t^w = \mathbf{e}_t^{\text{minute}} + \mathbf{e}_t^{\text{hour}} + \mathbf{e}_t^{\text{dow}}, \quad \mathbf{e}_t^m = \mathbf{e}_t^{\text{dom}} + \mathbf{e}_t^{\text{month}},$$

where $\mathbf{e}_t^w$ aggregates embeddings from features with week-level periodicity, and $\mathbf{e}_t^m$ aggregates those from features with longer month-level periodicity. Given a sequence $\mathbf{x}$, we first retrieve the corresponding temporal information based on its starting timestamp, and then integrate the retrieved information $\mathbf{e}_t^w$ and $\mathbf{e}_t^m$ into the representation as follow:

$$\mathbf{z} = \text{Linear}(\mathbf{x}), \quad \mathbf{z} \in \mathbb{R}^{d_{\text{model}}},$$

$$\mathbf{z}^t = \text{Concatenate}(\mathbf{z}, \mathbf{e}_t^w + \mathbf{e}_t^m), \quad \mathbf{z}^t \in \mathbb{R}^{d_{\text{model}} + T_{dim}}.$$

Here, $\mathbf{z}^t$ denotes the feature representation enriched with timestamp information, $d_{\text{model}}$ is the temporal representation dimension. In practice, a subset of these embeddings is selected according to the dataset characteristics. By default, $\mathbf{e}_t^w$ is adopted, as most datasets span sufficiently long periods to capture weekly patterns without severe overfitting. In contrast, $\mathbf{e}_t^m$ is usually excluded, since their monthly cycles are too long to provide enough samples for effective modeling in relatively small datasets. With this design, the TE module dynamically retrieves temporal priors from discrete calendar fields and integrates them into sequence representations, thereby enhancing the model's sensitivity to temporal variations.

## 3.4 CHANNEL EMBEDDING

To incorporate channel-specific identity information, the **Channel Embedding (CE)** module encodes variable indices into dedicated learnable embedding vectors. Given a multivariate input sequence $\mathbf{X} \in \mathbb{R}^{N \times L}$, we define a learnable channel embedding matrix $\mathbf{E}^{\text{identity}} \in \mathbb{R}^{N \times c_{\text{dim}}}$, where $c_{\text{dim}}$ ddenotes the embedding dimensionality. This matrix is initialized with zeros and jointly optimized during training.

For each variable $\mathbf{x}_n \in \{1, \ldots, N\}$, its corresponding identity embedding is retrieved by indexing into a learnable identity embedding matrix $\mathbf{E}^{\text{identity}} \in \mathbb{R}^{N \times C_{\text{dim}}}$. Specifically, the identity index is defined as $I_n = n - 1$, and the retrieved embedding is defined as:

$$\mathbf{e}_n^{\text{identity}} = \mathbf{E}^{\text{identity}}[I_n].$$

These identity embeddings are concatenated with the timestamp-enhanced sequence representations to form the input for subsequent encoding. Specifically, the timestamp-enhanced sequence $\mathbf{x}_n^{\text{ts}} \in \mathbb{R}^L$ is first projected into the model's feature space via a linear transformation and then concatenated with the identity embedding to yield the final representation:

$$\mathbf{z}_n^{tc} = \text{Concatenate}(\mathbf{z}_n^t, \mathbf{e}_n^{\text{identity}}), \quad \mathbf{z}_n^{tc} \in \mathbb{R}^{d_{\text{model}} + T_{\text{dim}} + C_{\text{dim}}},$$

where $\mathbf{z}_n^t$ denotes the projected timestamp-enhanced features, and $\mathbf{z}_n^{tc}$ represents the identity-enriched embedding that jointly encodes temporal dynamics and variable-specific semantics. With this design, the identity information of different channels can be effectively fused into their corresponding representations.

| Models | IndexNet | | SOFTS | | iTransformer | | PatchTST | | Crossformer | | TiDE | | TimesNet | | DLinear | | SCINet | | FEDformer | | TSMixer | | Autoformer | |
|---|---|---|---|---|---|---|---|---|---|---|---|---|---|---|---|---|---|---|---|---|---|---|---|---|
| Metric | MSE | MAE | MSE | MAE | MSE | MAE | MSE | MAE | MSE | MAE | MSE | MAE | MSE | MAE | MSE | MAE | MSE | MAE | MSE | MAE | MSE | MAE | MSE | MAE |
| ETTh1 | **0.448** | **0.436** | 0.449 | 0.442 | 0.454 | 0.447 | 0.469 | 0.454 | 0.529 | 0.522 | 0.541 | 0.507 | 0.458 | 0.450 | 0.456 | 0.452 | 0.747 | 0.647 | 0.440 | 0.460 | 0.463 | 0.452 | 0.496 | 0.512 |
| ETTh2 | 0.381 | 0.405 | **0.373** | **0.400** | 0.383 | 0.407 | 0.387 | 0.407 | 0.942 | 0.684 | 0.611 | 0.550 | 0.414 | 0.427 | 0.559 | 0.515 | 0.954 | 0.723 | 0.437 | 0.449 | 0.401 | 0.417 | 0.450 | 0.459 |
| ETTm1 | **0.374** | **0.392** | 0.393 | 0.403 | 0.407 | 0.410 | 0.387 | 0.400 | 0.513 | 0.496 | 0.419 | 0.419 | 0.400 | 0.406 | 0.403 | 0.407 | 0.485 | 0.481 | 0.448 | 0.452 | 0.398 | 0.407 | 0.588 | 0.517 |
| ETTm2 | **0.278** | **0.321** | 0.287 | 0.330 | 0.288 | 0.332 | 0.281 | 0.326 | 0.757 | 0.610 | 0.358 | 0.404 | 0.291 | 0.333 | 0.350 | 0.401 | 0.571 | 0.537 | 0.305 | 0.349 | 0.289 | 0.333 | 0.327 | 0.371 |
| Weather | **0.240** | **0.268** | 0.255 | 0.278 | 0.258 | 0.278 | 0.259 | 0.281 | 0.259 | 0.315 | 0.271 | 0.320 | 0.259 | 0.287 | 0.265 | 0.317 | 0.292 | 0.363 | 0.309 | 0.360 | 0.256 | 0.279 | 0.338 | 0.382 |
| ECL | **0.169** | **0.259** | 0.174 | 0.264 | 0.178 | 0.270 | 0.205 | 0.290 | 0.244 | 0.334 | 0.251 | 0.344 | 0.192 | 0.295 | 0.212 | 0.300 | 0.268 | 0.365 | 0.214 | 0.327 | 0.186 | 0.287 | 0.227 | 0.338 |
| Traffic | 0.411 | **0.267** | **0.409** | **0.267** | 0.428 | 0.282 | 0.481 | 0.304 | 0.550 | 0.304 | 0.760 | 0.473 | 0.620 | 0.336 | 0.625 | 0.383 | 0.804 | 0.509 | 0.610 | 0.376 | 0.522 | 0.357 | 0.628 | 0.379 |
| Solar-Energy | **0.223** | **0.256** | 0.229 | **0.256** | 0.233 | 0.262 | 0.270 | 0.307 | 0.641 | 0.639 | 0.347 | 0.417 | 0.301 | 0.319 | 0.330 | 0.401 | 0.282 | 0.375 | 0.291 | 0.381 | 0.260 | 0.297 | 0.885 | 0.711 |

Table 1: Multivariate forecasting results with prediction lengths $T \in \{96, 192, 336, 720\}$ for all datasets and fixed lookback length $L = 96$. Results are averaged from all prediction lengths. Full results are listed in Appendix B.6.1 and the short-term forecasting results on PeMS datasets are show in Appendix B.7.

## 3.5 FORECASTING

The Forecasting module consists of two primary components: an MLP-based encoder that processes the index-enriched sequence representations, and a linear projection layer that generates the final forecasts. The output of the CE module $\mathbf{Z}^{\mathbf{tc}} \in \mathbb{R}^{N \times (d_{\text{model}} + T_{\text{dim}} + C_{\text{dim}})}$ is fed into a stack of feedforward layers designed to capture complex variable-wise interactions and produce the forecasting outputs. Formally, the prediction process consists of $m$ residual MLP layers:

$$\mathbf{H}^{(0)} = \mathbf{Z}^{\mathbf{tc}}, \quad \mathbf{H}^{(l)} = \mathbf{H}^{(l-1)} + \text{Linear}(\text{ReLU}(\text{Linear}(\mathbf{H}^{(l-1)}))), \quad l = \{1, \ldots, m\},$$

where each layer consists of two linear transformations separated by a ReLU activation, with residual connections applied to facilitate gradient flow and stabilize training. The output of the last layer $\mathbf{H}^{(n)}$ is then passed through a final linear projection that maps the features to the output sequence length $T$:

$$\hat{\mathbf{Y}} = \text{Linear}(\mathbf{H}^{(n)}), \quad \hat{\mathbf{Y}} \in \mathbb{R}^{N \times T}.$$

Finally, we restore the predicted outputs to their original scale by de-normalization operation, generating final output $\mathbf{Y} \in \mathbb{R}^{N \times T}$.

## 4 EXPERIMENT

To validate the effectiveness of the proposed IndexNet, we conduct extensive experiments across various time series forecasting tasks, encompassing both long-term and short-term scenarios.

**Baselines.** We compare IndexNet against a diverse set of state-of-the-art models, including Transformer-based methods such as iTransformer (Liu et al., 2024c), PatchTST (Nie et al., 2023), Crossformer (Zhang & Yan, 2023), FEDformer (Zhou et al., 2022), Autoformer (Wu et al., 2021), and Informer (Zhou et al., 2021); MLP-based methods including SOFTS (Han et al., 2024), TiDE (Das et al., 2023), DLinear (Zeng et al., 2023), and TSMixer (Wang et al., 2024b); as well as CNN-based models such as TimesNet (Wu et al., 2023), SCINet (Liu et al., 2022), and MICN (Wang et al., 2022).

**Implementation Details.** All experiments are implemented in PyTorch (Paszke et al., 2019) and conducted on a single NVIDIA A100 80GB GPU. We adopt the Adam optimizer (Kingma, 2014). More implementation details about hyperparameter settings and metrics are provided in Appendix Sec. B.2 and Sec. B.3, respectively.

### 4.1 LONG-TERM FORECASTING

**Setups.** We conduct long-term forecasting experiments on several widely used real-world datasets, including the Electricity Transformer Temperature (ETT) dataset with its four subsets (ETTh1, ETTh2, ETTm1, ETTm2) (Wu et al., 2021; Miao et al., 2024a), as well as Weather, Electricity, Traffic, and Solar (Liu et al., 2025a;b). Following previous works (Zhou et al., 2021; Wu et al., 2021), we use Mean Squared Error (MSE) and Mean Absolute Error (MAE) as evaluation metrics. We set the input length $L$ to 96 for all methods. The detailed introduction can be found in B.1.

**Results.** As shown in Tab. 1, **IndexNet** consistently achieves either the best or second-best performance across most datasets, with the only exception being the Solar dataset. While our model falls slightly behind the top-performing method on a few individual datasets, it delivers overall superior results. Remarkably, despite adopting a relatively simple MLP-based backbone and a channel-independent (CI) modeling strategy—typically more effective in low-dimensional settings—IndexNet remains highly competitive even on the dataset with a large number of channels. Specifically, IndexNet outperforms state-of-the-art Transformer-based channel-dependent (CD) models such as iTransformer and Crossformer, reducing MSE by 2.34% and 24% on average, respectively. In contrast, on the lower-dimensional ETTm1 dataset, IndexNet also demonstrates strong performance, surpassing representative CI baselines like PatchTST and DLinear with MSE reductions of 8.11% and 6.5%, respectively. These results highlight the robustness and adaptability of IndexNet across datasets with varying dimensionality and dependency structures.

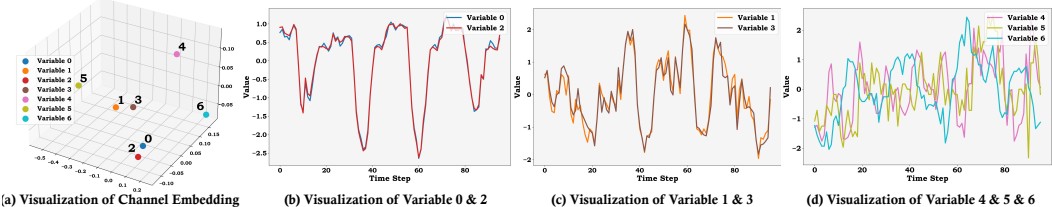

Figure 3: Visualization of Channel Embeddings and multivariate time series from the ETTh1 dataset. (a) shows the 3D projection of learned channel embeddings after dimensionality reduction via PCA. (b), (c), and (d) illustrate the time sequences of selected variable groups.

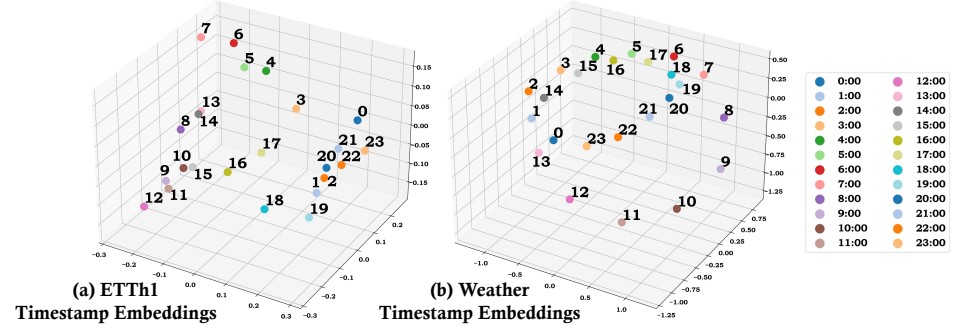

Figure 4: **Visualization of 24-hour Timestamp Embeddings.** Subfigure (a) displays the 3D PCA projection of Timestamp Embeddings on the ETTh1 dataset, while subfigure (b) shows the corresponding visualization on the Weather dataset.

## 4.2 ABLATION STUDIES

| Case | TE | CE | ETTm1 | | Weather | | Solar | | Electricity | | Traffic | |
|------|-----|-----|-------|-------|---------|-------|-------|-------|-------------|-------|---------|-------|
| | | | MSE | MAE | MSE | MAE | MSE | MAE | MSE | MAE | MSE | MAE |
| ① | × | × | 0.386 | 0.398 | 0.257 | 0.278 | 0.266 | 0.290 | 0.189 | 0.274 | 0.459 | 0.288 |
| ② | × | ✓ | 0.380 | 0.396 | 0.245 | 0.272 | 0.260 | 0.287 | 0.179 | 0.267 | 0.458 | 0.286 |
| ③ | ✓ | × | 0.381 | 0.396 | 0.252 | 0.274 | 0.238 | 0.269 | 0.178 | 0.266 | 0.415 | 0.272 |
| ④ | ✓ | ✓ | **0.374** | **0.392** | **0.240** | **0.268** | **0.223** | **0.256** | **0.169** | **0.259** | **0.411** | **0.267** |

Table 2: Ablation on the effect of removing TE and CE sub-modules. ✓ indicates the use of the sub-module, while × means removing the sub-module. Results are averaged from all prediction lengths. Full results are listed in Appendix B.8.

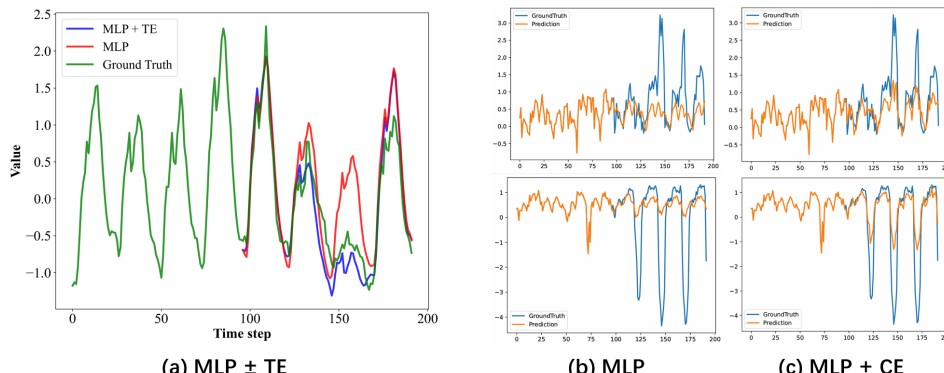

(a) MLP ± TE            (b) MLP            (c) MLP + CE

Figure 5: **Visualization of ablation results in Electricity dataset.** Subfigure (a) illustrates the difference in capturing week-level periodicity with and without the TE module, while (b) and (c) compare the predictions of two similar channels before and after introducing the CE module.

**Components Ablation.** To assess the effectiveness of the temporal and channel-aware enhancement modules, namely **TE** (Timestamp Embedding) and **CE** (Channel Embedding), we conduct ablation studies by selectively removing each component. As shown in Tab. 2, removing the TE module results in a clear drop in forecasting accuracy, especially on datasets with strong periodicity such as Solar, Electricity, and Traffic. More intuitively, Fig. 5(a) shows that with TE, our MLP model can not only leverage day-level variations (e.g., Monday-Thursday) to predict Friday and Monday, but also use week-level timestamp cues to distinguish the different patterns of Saturday and Sunday. Furthermore, Fig. 5(b) illustrates that with CE, the channel-independent model no longer produces overly similar predictions for correlated channels, but instead adjusts them according to the characteristics of each channel sequence.

| Methods | | IndexNet | | | | | PatchTST | | | | | iTransformer | | | |
|---|---|---|---|---|---|---|---|---|---|---|---|---|---|---|---|
| Metric | | MSE | MAE | Params(M) | GFLOPs | Test(ms) | MSE | MAE | Params(M) | GFLOPs | Temst(ms) | MSE | MAE | Params(M) | GFLOPs m | Test(ms) |
| Weather | 96 | **0.150** | **0.198** | **1.77** | **0.596** | **0.957** | 0.152 | 0.199 | 4.133 | 7.962 | 23.671 | 0.155 | 0.206 | 4.870 | 3.271 | 2.239 |
| | 192 | **0.192** | **0.241** | **1.828** | **0.613** | **0.962** | 0.197 | 0.243 | 8.199 | 8.483 | 23.871 | 0.200 | 0.245 | 4.920 | 3.304 | 2.245 |
| | 336 | **0.247** | **0.285** | **1.907** | **0.639** | **0.974** | 0.249 | 0.283 | 14.298 | 9.263 | 23.918 | 0.252 | 0.289 | 4.994 | 3.354 | 2.253 |
| | 720 | **0.321** | **0.335** | **2.116** | **0.710** | **0.984** | 0.320 | 0.335 | 30.564 | 11.344 | 24.396 | 0.323 | 0.340 | 5.191 | 3.486 | 2.259 |
| Electricity | 96 | **0.125** | **0.218** | **1.776** | **9.104** | **2.385** | 0.130 | 0.222 | 62.222 | 30.427 | 90.367 | 0.132 | 0.230 | 5.465 | 28.052 | 9.143 |
| | 192 | **0.146** | **0.252** | **1.828** | **9.372** | **2.466** | 0.148 | 0.240 | 124.378 | 32.415 | 91.156 | 0.153 | 0.252 | 5.517 | 28.320 | 9.264 |
| | 336 | **0.163** | **0.255** | **1.907** | **9.774** | **2.572** | 0.167 | 0.261 | 217.612 | 35.397 | 91.896 | 0.167 | 0.264 | 5.595 | 28.723 | 9.377 |
| | 720 | **0.196** | **0.286** | **2.116** | **10.847** | **2.765** | 0.202 | 0.291 | 466.236 | 43.349 | 110.241 | 0.199 | 0.288 | 5.805 | 29.796 | 9.741 |
| Traffic | 96 | **0.334** | **0.240** | **6.446** | **88.805** | **15.683** | 0.367 | 0.251 | 500.931 | 327.539 | 298.311 | 0.354 | 0.256 | 13.326 | 183.727 | 61.380 |
| | 192 | **0.352** | **0.251** | **6.544** | **90.161** | **15.924** | 0.385 | 0.259 | 1001.498 | 339.551 | 300.224 | 0.369 | 0.269 | 13.424 | 184.213 | 61.934 |
| | 336 | **0.372** | **0.358** | **6.691** | **92.195** | **16.331** | 0.398 | 0.265 | 1752.348 | 357.568 | 303.281 | 0.389 | 0.271 | 13.514 | 186.322 | 62.229 |
| | 720 | **0.422** | **0.279** | **7.085** | **97.618** | **17.254** | 0.434 | 0.287 | 3754.615 | 405.615 | 312.592 | 0.435 | 0.301 | 13.815 | 190.474 | 63.176 |

Table 3: Comparison of performance and computational cost under the longer look-back window L=336.

**Longer Look-back Window and Computational Cost.** To evaluate the performance and computational efficiency of our method under longer input horizons, we conducted experiments with an extended look-back window of $L = 336$. The inference time was measured by averaging over 100 runs to estimate the cost of a single forward pass. As shown in Tab. 3, our method consistently achieves competitive or superior forecasting accuracy while maintaining significantly lower computational overhead compared with recent representative baselines.

Overall, the results demonstrate that our approach achieves a favorable balance between accuracy and efficiency. In terms of parameter size, IndexNet requires only 1.9M parameters, which is substantially fewer than PatchTST (over 14M) and iTransformer (approximately 5M). Regarding computational cost, IndexNet incurs merely 0.639 GFLOPs, compared with 9.263 GFLOPs for PatchTST and 3.354 GFLOPs for iTransformer. This lightweight design directly translates to faster inference: the average test latency of IndexNet is nearly 1 ms, far outperforming PatchTST (23.9 ms) and iTransformer (2.3 ms). These results clearly indicate that our method not only surpasses strong baselines in predictive performance, but also provides a substantial advantage in computational efficiency and inference speed, making it highly practical for real-world applications with limited resources.

| Models | iTransformer | | + IE | | + TE | | + CE | | + GALF | | + VH | | ModernTCN | | + IE | | + TE | | + CE | | + GALF | | + VH | |
|---|---|---|---|---|---|---|---|---|---|---|---|---|---|---|---|---|---|---|---|---|---|---|---|---|
| Metric | MSE | MAE | MSE | MAE | MSE | MAE | MSE | MAE | MSE | MAE | MSE | MAE | MSE | MAE | MSE | MAE | MSE | MAE | MSE | MAE | MSE | MAE | MSE | MAE |
| Weather 96 | 0.174 | 0.214 | **0.160** | **0.204** | 0.173 | 0.214 | 0.166 | 0.210 | 0.177 | 0.216 | 0.170 | 0.213 | 0.154 | 0.205 | **0.146** | **0.197** | 0.151 | 0.202 | 0.147 | 0.198 | 0.157 | 0.208 | 0.150 | 0.200 |
| Weather 192 | 0.221 | 0.254 | **0.210** | **0.250** | 0.224 | 0.256 | 0.212 | 0.251 | 0.224 | 0.259 | 0.219 | 0.252 | 0.203 | 0.246 | **0.193** | **0.241** | 0.199 | 0.245 | 0.194 | **0.241** | 0.205 | 0.249 | 0.198 | 0.244 |
| Weather 336 | 0.278 | 0.296 | **0.270** | **0.293** | 0.280 | 0.298 | **0.270** | 0.294 | 0.284 | 0.301 | 0.276 | 0.295 | 0.252 | 0.285 | **0.245** | **0.281** | 0.253 | 0.283 | 0.246 | **0.281** | 0.254 | 0.288 | 0.248 | 0.282 |
| Weather 720 | 0.358 | 0.347 | **0.350** | **0.345** | 0.356 | 0.348 | **0.350** | 0.346 | 0.372 | 0.358 | 0.357 | 0.347 | 0.318 | 0.334 | **0.308** | **0.329** | 0.316 | 0.333 | 0.311 | 0.331 | 0.322 | 0.336 | 0.315 | 0.329 |
| ECL 96 | 0.148 | 0.240 | **0.138** | **0.233** | 0.147 | 0.239 | 0.139 | **0.233** | 0.146 | 0.242 | 0.142 | 0.237 | 0.132 | 0.223 | **0.126** | **0.218** | 0.129 | 0.221 | 0.128 | 0.219 | 0.130 | 0.220 | 0.128 | 0.219 |
| ECL 192 | 0.162 | 0.253 | **0.154** | **0.247** | 0.160 | 0.251 | **0.154** | 0.248 | 0.163 | 0.253 | 0.159 | 0.251 | 0.149 | 0.242 | 0.143 | **0.237** | 0.144 | 0.238 | 0.144 | 0.238 | 0.146 | 0.240 | **0.142** | **0.237** |
| ECL 336 | 0.178 | 0.269 | **0.168** | **0.264** | 0.174 | 0.268 | 0.172 | 0.267 | 0.176 | 0.267 | 0.174 | 0.266 | 0.168 | 0.261 | **0.161** | **0.253** | 0.163 | 0.255 | 0.163 | 0.254 | 0.167 | 0.261 | 0.162 | 0.254 |
| ECL 720 | 0.225 | 0.317 | 0.202 | 0.296 | 0.235 | 0.318 | **0.201** | **0.295** | 0.221 | 0.314 | 0.211 | 0.302 | 0.204 | 0.294 | **0.196** | **0.286** | 0.197 | 0.287 | 0.199 | 0.288 | 0.202 | 0.293 | 0.198 | 0.288 |
| Traffic 96 | 0.395 | 0.268 | 0.391 | 0.265 | **0.380** | **0.266** | 0.428 | 0.279 | 0.393 | 0.270 | 0.392 | 0.266 | 0.369 | 0.255 | 0.341 | **0.242** | 0.343 | 0.245 | **0.340** | **0.242** | 0.372 | 0.258 | 0.366 | 0.252 |
| Traffic 192 | 0.417 | 0.276 | 0.421 | 0.275 | **0.391** | **0.274** | 0.451 | 0.287 | 0.415 | 0.277 | 0.415 | 0.276 | 0.388 | 0.264 | 0.362 | **0.248** | 0.366 | 0.250 | 0.363 | **0.248** | 0.385 | 0.255 | 0.385 | 0.252 |
| Traffic 336 | 0.392 | 0.283 | 0.434 | 0.283 | **0.408** | **0.281** | 0.465 | 0.292 | 0.431 | 0.284 | 0.430 | 0.282 | 0.397 | 0.265 | **0.380** | **0.250** | 0.384 | 0.253 | 0.382 | 0.251 | 0.393 | 0.261 | 0.387 | 0.258 |
| Traffic 720 | 0.467 | 0.302 | 0.469 | 0.293 | **0.437** | **0.292** | 0.508 | 0.309 | 0.462 | 0.300 | 0.458 | 0.298 | 0.438 | 0.288 | **0.414** | **0.277** | 0.416 | 0.280 | **0.414** | 0.278 | 0.430 | 0.285 | 0.419 | 0.280 |

Table 4: Applying Our Embedding Module to Different Network Architectures and Comparing with Existing Embedding Methods.

**Plug-and-Play.** To compare our method with existing embedding strategies across different network architectures, we conducted experiments on both the Transformer architecture (iTransformer (Liu et al., 2024c)) and the CNN architecture (ModernTCN (Donghao & Xue, 2024)). For the Transformer-based model, we followed the original hyperparameter settings, while for the CNN-based model, we directly replaced the FFN module with ModernTCN block and evaluated under the look-back window of length 336.

The results in Tab. 4 show that, in most cases, the proposed IE method effectively combines the performance gains brought by both TE and CE, and outperforms other existing embedding approaches (Yang et al., 2025; Wang et al., 2024a). For example, in the channel-independent CNN architecture, the CE module provides a substantial boost in accuracy, while in channel-dependent methods, the TE module demonstrates clear advantages. However, it is worth noting that when the IE module was applied to the Transformer-based model on the Traffic dataset, we observed obvious performance drop. We conjecture that this may be attributed to the intrinsic strength of iTransformer in capturing channel-specific dependencies, where the additional CE component could introduce redundancy, thereby diminishing overall predictive performance.

**Visualization Analysis.** We apply PCA to reduce the dimensions of **feature in latent space**, visualizing what the model has learned in Timestamp and Channel Embeddings. The results reveal clear temporal periodicity and inter-variable relationships. By examining spatial distances among channel embeddings, we observe the similarity of dynamic patterns across variables. As shown in Figure 3, variables 1 and 3, as well as 0 and 2, exhibit highly similar identity embeddings, matching their similar temporal behaviors in the raw sequences. In contrast, variables 4, 5, and 6 form a separate cluster, indicating a distinct group. Likewise, the learned timestamp embeddings capture periodic patterns. In (Figure 4(a)), the ETTh1 dataset—collected in China. We observe consistent electricity usage from 9 a.m. to 5 p.m., shifting toward evening routines, late-night consumption, and a unique dip around 1–2 p.m., reflecting the midday rest hours of Chinese people. Weather dataset embeddings reveal strong 24-hour periodicity, consistent with natural climate cycles. These results demonstrate the strong interpretability of our method, which is crucial for practical MTSF but has been largely overlooked.

## 5 CONCLUSION

In this work, we propose **IndexNet**, a simple yet effective MLP-based framework for multivariate time series forecasting. Unlike most existing methods, IndexNet explicitly incorporates index-related prior knowledge through a dedicated **Index Embedding (IE)** module. By integrating timestamp and variable index information via the proposed TE and CE components, IndexNet significantly enhances the model's temporal awareness and its ability to distinguish variable-specific patterns. This design enables more reliable forecasting and improves the interpretability of predictions—two critical yet often underexplored aspects in current MTSF research. Extensive experiments across diverse real-world datasets validate the effectiveness of our approach, highlighting the potential of incorporating index semantics in building temporally- and variably-aware forecasting models. We discuss the limitations and potential impacts of our work in Sec. A.

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

LLM USABLE STATEMENT

This paper employs LLMs solely for grammar checking and linguistic refinement, not involving any substantive content generation or fabrication.

## A  LIMITATIONS

Despite achieving significant progress on relatively simple linear-, Transformer-, and CNN-based architectures, our approach still faces challenges when extended to more complex models. While this work partially overcomes the limitation of current general forecasting methods that rely solely on input sequences, and provides a reliable, interpretable, and lightweight solution, integrating index information into sophisticated architectures remains non-trivial. In particular, given the fragility and intricate design of many existing models, how to avoid overfitting or disrupting the original structure when injecting index cues warrants further investigation. We believe these limitations can suggest promising directions for future research, including the development of more universal, architecture-agnostic schemes that can flexibly adapt to diverse scenarios.

## B  ADDITIONAL EXPERIMENT

### B.1  DESCRIPTION ON DATASETS

| Tasks | Dataset | Dim | Prediction Length | Dataset Size | Frequency | ADF$^\dagger$ | Timestamp |
|---|---|---|---|---|---|---|---|
| | ETTm1 | 7 | $\{96, 192, 336, 720\}$ | $(34465, 11521, 11521)$ | 15 min | $-14.98$ | ✓ |
| | ETTm2 | 7 | $\{96, 192, 336, 720\}$ | $(34465, 11521, 11521)$ | 15 min | $-5.66$ | ✓ |
| | ETTh1 | 7 | $\{96, 192, 336, 720\}$ | $(8545, 2881, 2881)$ | 1 hour | $-5.91$ | ✓ |
| Long-term | ETTh2 | 7 | $\{96, 192, 336, 720\}$ | $(8545, 2881, 2881)$ | 1 hour | $-4.13$ | ✓ |
| Forecasting | Electricity | 321 | $\{96, 192, 336, 720\}$ | $(18317, 2633, 5261)$ | 1 hour | $-8.44$ | ✓ |
| | Traffic | 862 | $\{96, 192, 336, 720\}$ | $(12185, 1757, 3509)$ | 1 hour | $-15.02$ | ✓ |
| | Weather | 21 | $\{96, 192, 336, 720\}$ | $(36792, 5271, 10540)$ | 10 min | $-26.68$ | ✓ |
| | Solar-Energy | 137 | $\{96, 192, 336, 720\}$ | $(36601, 5161, 10417)$ | 10 min | $-37.23$ | × |
| | PeMS03 | 358 | 12 | $(15617, 5135, 5135)$ | 5 min | $-19.05$ | × |
| Short-term | PeMS04 | 307 | 12 | $(10172, 3375, 3375)$ | 5 min | $-15.66$ | × |
| Forecasting | PeMS07 | 883 | 12 | $(16911, 5622, 5622)$ | 5 min | $-20.60$ | × |
| | PeMS08 | 170 | 12 | $(10690, 3548, 265)$ | 5 min | $-16.04$ | × |

Timestamp: the ✓ means the dataset contains explicit timestamp, and × is not.
$\dagger$ Augmented Dickey-Fuller (ADF) Test: A smaller ADF test result indicates a more stationary time series data.

Table 5: Dataset detailed descriptions. "Dataset Size" denotes the total number of time points in (Train(s), Validation, Test) split respectively. "Prediction Length" denotes the future time points to be predicted. "Frequency" denotes the sampling interval of time points.

We conduct extensive experiments on several widely-used time series datasets for long-term forecasting. Additionally, we use the PeMS datasets for short-term forecasting. We report the statistics in Tab. 5. Detailed descriptions of these datasets are as follows:

(1) **ETT** (Electricity Transformer Temperature) dataset (Zhou et al., 2021) encompasses temperature and power load data from electricity transformers in two regions of China, spanning from 2016 to 2018. This dataset has two granularity levels: ETTh (hourly) and ETTm (15 minutes).

(2) **Weather** dataset (Wu et al., 2023) captures 21 distinct meteorological indicators in Germany, meticulously recorded at 10-minute intervals throughout 2020. Key indicators in this dataset include air temperature, visibility, among others, offering a comprehensive view of the weather dynamics.

(3) **Electricity** dataset (Wu et al., 2023) features hourly electricity consumption records in kilowatt-hours (kWh) for 321 clients. Sourced from the UCL Machine Learning Repository, this dataset covers the period from 2012 to 2014, providing valuable insights into consumer electricity usage patterns.

(4) **Traffic** dataset (Wu et al., 2023) includes data on hourly road occupancy rates, gathered by 862 detectors across the freeways of the San Francisco Bay area. This dataset, covering the years 2015 to 2016, offers a detailed snapshot of traffic flow and congestion.

(5) **Solar-Energy** dataset (Liu et al., 2024c) contains solar power production data recorded every 10 minutes throughout 2006 from 137 photovoltaic (PV) plants in Alabama.

(6) **PeMS** dataset (Liu et al., 2022) comprises four public traffic network datasets (PeMS03, PeMS04, PeMS07, and PeMS08), constructed from the Caltrans Performance Measurement System (PeMS) across four districts in California. The data is aggregated into 5-minute intervals, resulting in 12 data points per hour and 288 data points per day.

## B.2 HYPERPARAMETERS SETTINGS

| | $n$ layers | $T_{dim}$ | $C_{dim}$ | Timestamp | lr | d_model | d_ff |
|---|---|---|---|---|---|---|---|
| ETTh1 | 3 | 16 | 16 | ✓ | 5e-4 | 128 | 128 |
| ETTh2 | 2 | 16 | 16 | ✓ | 5e-5 | 128 | 128 |
| ETTm1 | 3 | 16 | 16 | ✓ | 2e-4 | 128 | 128 |
| ETTm2 | 3 | 16 | 16 | ✓ | 2e-4 | 128 | 128 |
| Weather | 3 | 16 | 16 | ✓ | 5e-4 | 512 | 512 |
| Solar | 2 | 16 | 16 | × | 5e-4 | 512 | 512 |
| Electricity | 3 | 16 | 16 | ✓ | 1e-3 | 512 | 512 |
| Traffic | 3 | 256 | 256 | ✓ | 1e-3 | 512 | 1024 |
| PSME03 | 3 | 16 | 16 | × | 1e-3 | 512 | 512 |
| PSME04 | 3 | 16 | 16 | × | 1e-3 | 512 | 512 |
| PSME07 | 3 | 16 | 16 | × | 1e-3 | 512 | 512 |
| PSME08 | 3 | 16 | 16 | × | 1e-3 | 512 | 512 |

Table 6: Hyperparameter settings for different datasets. In fixed timestamp column, the ✓ means we only use minute of hour, hour of day, and day of week, while the × means we additionally use the year and the month of year. It is notably that we only use it in traffic dataset, because pervious work (Das et al., 2023) has found that traffic dataset can be benefited from the calendar information for the complete year.

## B.3 PERFORMANCE METRICS

### B.4 LONG-TERM FORECASTING

We use Mean Squared Error (MSE) and Mean Absolute Error (MAE) as evaluation metrics. Given the ground truth values $\mathbf{X}_i$ and the predicted values $\hat{\mathbf{X}}_i$, these metrics are defined as follows:

$$\text{MSE} = \frac{1}{N} \sum_{i=1}^{N} (\mathbf{X}_i - \hat{\mathbf{X}}_i)^2, \quad \text{MAE} = \frac{1}{N} \sum_{i=1}^{N} |\mathbf{X}_i - \hat{\mathbf{X}}_i|,$$

where $N$ is the total number of predictions.

### B.5 SHORT-TERM FORECASTING

We use MAE (the same as defined above), Mean Absolute Percentage Error (MAPE), and Root Mean Squared Error (RMSE) to evaluate the performance. These metrics are defined as follows:

$$\text{MAPE} = \frac{1}{N} \sum_{i=1}^{N} \left| \frac{\mathbf{X}_i - \hat{\mathbf{X}}_i}{\mathbf{X}_i} \right| \times 100, \quad \text{RMSE} = \sqrt{\frac{1}{N} \sum_{i=1}^{N} (\mathbf{X}_i - \hat{\mathbf{X}}_i)^2}.$$

## B.6 FULL RESULTS

### B.6.1 LONG-TERM FORECASTING

Table 7: Full results of the long-term forecasting task. We compare extensive competitive models under different prediction lengths following the setting of TimesNet (Wu et al., 2023). The input sequence length is set to 96 for all baselines. *Avg* means the average results from all four prediction lengths.

| Models | | IndexNet | | SOFTS | | iTransformer | | PatchTST | | Crossformer | | TiDE | | TimesNet | | DLinear | | SCINet | | FEDformer | | TSMixer | | Autoformer | |
|---|---|---|---|---|---|---|---|---|---|---|---|---|---|---|---|---|---|---|---|---|---|---|---|---|---|
| Metric | | MSE | MAE | MSE | MAE | MSE | MAE | MSE | MAE | MSE | MAE | MSE | MAE | MSE | MAE | MSE | MAE | MSE | MAE | MSE | MAE | MSE | MAE | MSE | MAE |
| ETTm1 | 96 | 0.312 | 0.352 | 0.325 | 0.361 | 0.334 | 0.368 | 0.329 | 0.367 | 0.404 | 0.426 | 0.364 | 0.387 | 0.338 | 0.375 | 0.345 | 0.372 | 0.418 | 0.438 | 0.379 | 0.419 | 0.323 | 0.363 | 0.505 | 0.475 |
| | 192 | 0.355 | 0.379 | 0.375 | 0.389 | 0.377 | 0.391 | 0.367 | 0.385 | 0.450 | 0.451 | 0.398 | 0.404 | 0.374 | 0.387 | 0.380 | 0.389 | 0.439 | 0.450 | 0.426 | 0.441 | 0.376 | 0.392 | 0.553 | 0.496 |
| | 336 | 0.386 | 0.402 | 0.405 | 0.412 | 0.426 | 0.420 | 0.399 | 0.410 | 0.532 | 0.515 | 0.428 | 0.425 | 0.410 | 0.411 | 0.413 | 0.413 | 0.490 | 0.485 | 0.445 | 0.459 | 0.407 | 0.413 | 0.621 | 0.537 |
| | 720 | 0.445 | 0.437 | 0.466 | 0.447 | 0.491 | 0.459 | 0.454 | 0.439 | 0.666 | 0.589 | 0.487 | 0.461 | 0.478 | 0.450 | 0.474 | 0.453 | 0.595 | 0.550 | 0.543 | 0.490 | 0.485 | 0.459 | 0.671 | 0.561 |
| | Avg | 0.374 | 0.392 | 0.393 | 0.403 | 0.407 | 0.410 | 0.387 | 0.400 | 0.513 | 0.496 | 0.419 | 0.419 | 0.400 | 0.406 | 0.403 | 0.407 | 0.485 | 0.481 | 0.448 | 0.452 | 0.398 | 0.407 | 0.588 | 0.517 |
| ETTm2 | 96 | 0.173 | 0.255 | 0.180 | 0.261 | 0.180 | 0.264 | 0.175 | 0.259 | 0.287 | 0.366 | 0.207 | 0.305 | 0.187 | 0.267 | 0.193 | 0.292 | 0.286 | 0.377 | 0.203 | 0.287 | 0.182 | 0.266 | 0.255 | 0.339 |
| | 192 | 0.240 | 0.297 | 0.246 | 0.306 | 0.250 | 0.309 | 0.241 | 0.302 | 0.414 | 0.492 | 0.290 | 0.364 | 0.249 | 0.309 | 0.284 | 0.362 | 0.399 | 0.445 | 0.269 | 0.328 | 0.249 | 0.309 | 0.281 | 0.340 |
| | 336 | 0.301 | 0.338 | 0.319 | 0.352 | 0.311 | 0.348 | 0.305 | 0.343 | 0.597 | 0.542 | 0.377 | 0.422 | 0.321 | 0.351 | 0.369 | 0.427 | 0.637 | 0.591 | 0.325 | 0.366 | 0.309 | 0.347 | 0.339 | 0.372 |
| | 720 | 0.398 | 0.395 | 0.405 | 0.401 | 0.412 | 0.407 | 0.402 | 0.400 | 1.730 | 1.042 | 0.558 | 0.524 | 0.408 | 0.403 | 0.554 | 0.522 | 0.960 | 0.735 | 0.421 | 0.415 | 0.416 | 0.408 | 0.433 | 0.432 |
| | Avg | 0.278 | 0.321 | 0.287 | 0.330 | 0.288 | 0.332 | 0.281 | 0.326 | 0.757 | 0.610 | 0.358 | 0.404 | 0.291 | 0.333 | 0.350 | 0.401 | 0.571 | 0.537 | 0.305 | 0.349 | 0.289 | 0.333 | 0.327 | 0.371 |
| ETTh1 | 96 | 0.378 | 0.393 | 0.381 | 0.399 | 0.386 | 0.405 | 0.414 | 0.419 | 0.423 | 0.448 | 0.479 | 0.464 | 0.384 | 0.402 | 0.386 | 0.400 | 0.654 | 0.599 | 0.376 | 0.419 | 0.401 | 0.412 | 0.449 | 0.459 |
| | 192 | 0.435 | 0.425 | 0.435 | 0.431 | 0.441 | 0.436 | 0.460 | 0.445 | 0.471 | 0.474 | 0.525 | 0.492 | 0.436 | 0.429 | 0.437 | 0.432 | 0.719 | 0.631 | 0.420 | 0.448 | 0.452 | 0.442 | 0.500 | 0.482 |
| | 336 | 0.483 | 0.450 | 0.480 | 0.452 | 0.487 | 0.458 | 0.501 | 0.466 | 0.570 | 0.546 | 0.565 | 0.515 | 0.491 | 0.469 | 0.481 | 0.459 | 0.778 | 0.659 | 0.459 | 0.465 | 0.492 | 0.463 | 0.521 | 0.496 |
| | 720 | 0.495 | 0.475 | 0.499 | 0.488 | 0.503 | 0.491 | 0.500 | 0.488 | 0.653 | 0.621 | 0.594 | 0.558 | 0.521 | 0.500 | 0.519 | 0.516 | 0.836 | 0.699 | 0.506 | 0.507 | 0.507 | 0.490 | 0.514 | 0.512 |
| | Avg | 0.448 | 0.436 | 0.449 | 0.442 | 0.454 | 0.447 | 0.469 | 0.454 | 0.529 | 0.522 | 0.541 | 0.507 | 0.458 | 0.450 | 0.456 | 0.452 | 0.747 | 0.647 | 0.440 | 0.460 | 0.463 | 0.452 | 0.496 | 0.487 |
| ETTh2 | 96 | 0.295 | 0.345 | 0.297 | 0.347 | 0.297 | 0.349 | 0.302 | 0.348 | 0.745 | 0.584 | 0.400 | 0.440 | 0.340 | 0.374 | 0.333 | 0.387 | 0.707 | 0.621 | 0.358 | 0.397 | 0.319 | 0.361 | 0.346 | 0.388 |
| | 192 | 0.377 | 0.395 | 0.373 | 0.394 | 0.380 | 0.400 | 0.388 | 0.400 | 0.877 | 0.656 | 0.528 | 0.509 | 0.402 | 0.414 | 0.477 | 0.476 | 0.860 | 0.689 | 0.429 | 0.439 | 0.402 | 0.410 | 0.456 | 0.452 |
| | 336 | 0.419 | 0.431 | 0.410 | 0.426 | 0.428 | 0.432 | 0.426 | 0.433 | 1.043 | 0.731 | 0.643 | 0.571 | 0.452 | 0.452 | 0.594 | 0.541 | 1.000 | 0.744 | 0.496 | 0.487 | 0.444 | 0.446 | 0.482 | 0.486 |
| | 720 | 0.431 | 0.447 | 0.411 | 0.433 | 0.427 | 0.445 | 0.431 | 0.446 | 1.104 | 0.763 | 0.874 | 0.679 | 0.462 | 0.468 | 0.831 | 0.657 | 1.249 | 0.838 | 0.463 | 0.474 | 0.441 | 0.450 | 0.515 | 0.511 |
| | Avg | 0.381 | 0.405 | 0.373 | 0.400 | 0.383 | 0.407 | 0.387 | 0.407 | 0.942 | 0.684 | 0.611 | 0.550 | 0.414 | 0.427 | 0.559 | 0.515 | 0.954 | 0.723 | 0.437 | 0.449 | 0.401 | 0.417 | 0.450 | 0.459 |
| ECL | 96 | 0.137 | 0.230 | 0.143 | 0.233 | 0.148 | 0.240 | 0.181 | 0.270 | 0.219 | 0.314 | 0.237 | 0.329 | 0.168 | 0.272 | 0.197 | 0.282 | 0.247 | 0.345 | 0.193 | 0.308 | 0.157 | 0.260 | 0.201 | 0.317 |
| | 192 | 0.154 | 0.245 | 0.158 | 0.248 | 0.162 | 0.253 | 0.188 | 0.274 | 0.231 | 0.322 | 0.236 | 0.330 | 0.184 | 0.289 | 0.196 | 0.285 | 0.257 | 0.355 | 0.201 | 0.315 | 0.173 | 0.274 | 0.222 | 0.334 |
| | 336 | 0.171 | 0.262 | 0.178 | 0.269 | 0.178 | 0.269 | 0.204 | 0.293 | 0.246 | 0.337 | 0.249 | 0.344 | 0.198 | 0.300 | 0.209 | 0.301 | 0.269 | 0.369 | 0.214 | 0.329 | 0.192 | 0.295 | 0.231 | 0.338 |
| | 720 | 0.212 | 0.299 | 0.218 | 0.305 | 0.225 | 0.317 | 0.246 | 0.324 | 0.280 | 0.363 | 0.284 | 0.373 | 0.220 | 0.320 | 0.245 | 0.333 | 0.299 | 0.390 | 0.246 | 0.355 | 0.223 | 0.318 | 0.254 | 0.361 |
| | Avg | 0.169 | 0.259 | 0.174 | 0.264 | 0.178 | 0.270 | 0.205 | 0.290 | 0.244 | 0.334 | 0.251 | 0.344 | 0.192 | 0.295 | 0.212 | 0.300 | 0.268 | 0.365 | 0.214 | 0.327 | 0.186 | 0.287 | 0.227 | 0.338 |
| Traffic | 96 | 0.384 | 0.253 | 0.376 | 0.251 | 0.395 | 0.268 | 0.462 | 0.295 | 0.522 | 0.290 | 0.805 | 0.493 | 0.593 | 0.321 | 0.650 | 0.396 | 0.788 | 0.499 | 0.587 | 0.366 | 0.493 | 0.336 | 0.613 | 0.388 |
| | 192 | 0.391 | 0.260 | 0.398 | 0.361 | 0.417 | 0.276 | 0.466 | 0.296 | 0.530 | 0.293 | 0.756 | 0.474 | 0.617 | 0.336 | 0.598 | 0.370 | 0.789 | 0.505 | 0.604 | 0.373 | 0.497 | 0.351 | 0.616 | 0.382 |
| | 336 | 0.411 | 0.270 | 0.415 | 0.269 | 0.433 | 0.283 | 0.482 | 0.304 | 0.558 | 0.305 | 0.762 | 0.477 | 0.629 | 0.336 | 0.605 | 0.373 | 0.797 | 0.508 | 0.621 | 0.383 | 0.528 | 0.328 | 0.361 | 0.337 |
| | 720 | 0.459 | 0.286 | 0.447 | 0.287 | 0.467 | 0.302 | 0.514 | 0.322 | 0.589 | 0.328 | 0.719 | 0.449 | 0.640 | 0.350 | 0.645 | 0.394 | 0.841 | 0.523 | 0.626 | 0.382 | 0.569 | 0.380 | 0.660 | 0.408 |
| | Avg | 0.411 | 0.267 | 0.409 | 0.267 | 0.428 | 0.282 | 0.481 | 0.304 | 0.550 | 0.304 | 0.760 | 0.473 | 0.620 | 0.336 | 0.625 | 0.383 | 0.804 | 0.509 | 0.610 | 0.376 | 0.522 | 0.357 | 0.628 | 0.379 |
| Weather | 96 | 0.155 | 0.198 | 0.166 | 0.208 | 0.174 | 0.214 | 0.177 | 0.218 | 0.158 | 0.230 | 0.202 | 0.261 | 0.172 | 0.220 | 0.196 | 0.255 | 0.221 | 0.306 | 0.217 | 0.296 | 0.166 | 0.210 | 0.266 | 0.336 |
| | 192 | 0.202 | 0.243 | 0.217 | 0.253 | 0.221 | 0.254 | 0.225 | 0.259 | 0.206 | 0.277 | 0.242 | 0.298 | 0.219 | 0.261 | 0.237 | 0.296 | 0.261 | 0.340 | 0.276 | 0.336 | 0.215 | 0.256 | 0.307 | 0.367 |
| | 336 | 0.260 | 0.287 | 0.282 | 0.300 | 0.278 | 0.296 | 0.278 | 0.297 | 0.272 | 0.335 | 0.287 | 0.335 | 0.280 | 0.306 | 0.283 | 0.335 | 0.309 | 0.378 | 0.339 | 0.380 | 0.287 | 0.300 | 0.359 | 0.395 |
| | 720 | 0.343 | 0.341 | 0.356 | 0.351 | 0.358 | 0.347 | 0.354 | 0.348 | 0.398 | 0.418 | 0.351 | 0.323 | 0.365 | 0.359 | 0.345 | 0.381 | 0.377 | 0.427 | 0.403 | 0.428 | 0.355 | 0.348 | 0.419 | 0.428 |
| | Avg | 0.240 | 0.268 | 0.255 | 0.278 | 0.258 | 0.278 | 0.259 | 0.281 | 0.259 | 0.315 | 0.271 | 0.320 | 0.259 | 0.287 | 0.265 | 0.317 | 0.292 | 0.363 | 0.309 | 0.360 | 0.256 | 0.279 | 0.338 | 0.382 |
| Solar-Energy | 96 | 0.192 | 0.230 | 0.200 | 0.230 | 0.203 | 0.237 | 0.234 | 0.286 | 0.310 | 0.331 | 0.312 | 0.399 | 0.250 | 0.292 | 0.290 | 0.378 | 0.237 | 0.344 | 0.242 | 0.342 | 0.221 | 0.275 | 0.884 | 0.711 |
| | 192 | 0.219 | 0.255 | 0.229 | 0.253 | 0.233 | 0.261 | 0.267 | 0.310 | 0.734 | 0.725 | 0.339 | 0.416 | 0.296 | 0.318 | 0.320 | 0.398 | 0.280 | 0.380 | 0.285 | 0.380 | 0.268 | 0.306 | 0.834 | 0.692 |
| | 336 | 0.239 | 0.271 | 0.243 | 0.269 | 0.248 | 0.273 | 0.290 | 0.315 | 0.750 | 0.735 | 0.368 | 0.430 | 0.319 | 0.330 | 0.353 | 0.415 | 0.304 | 0.389 | 0.282 | 0.376 | 0.272 | 0.294 | 0.941 | 0.723 |
| | 720 | 0.241 | 0.269 | 0.245 | 0.272 | 0.249 | 0.275 | 0.289 | 0.317 | 0.769 | 0.765 | 0.370 | 0.425 | 0.338 | 0.337 | 0.356 | 0.413 | 0.308 | 0.388 | 0.357 | 0.427 | 0.281 | 0.313 | 0.882 | 0.717 |
| | Avg | 0.223 | 0.256 | 0.229 | 0.256 | 0.233 | 0.262 | 0.270 | 0.307 | 0.641 | 0.639 | 0.347 | 0.416 | 0.301 | 0.319 | 0.330 | 0.401 | 0.282 | 0.375 | 0.291 | 0.381 | 0.260 | 0.297 | 0.885 | 0.711 |

## B.7 SHORT-TERM FORECASTING

**Setups.** For short-term forecasting, we conduct experiments on PeMS datasets (Wang et al., 2024b), which capture complex spatio-temporal correlations among multiple variates across city-wide traffic networks. We use mean absolute error (MAE), mean absolute percentage error (MAPE), and root mean squared error (RMSE) as evaluation metrics. The input length $L$ is set to 96 and the output length $T$ to 12 for all baselines. Details of datasets and metrics are in Sec. B.1 and Sec. B.5.

**Results** As shown in Table Tab. 8, methods that perform well in long-term forecasting under channel-independent (CI) settings, such as PatchTST (Nie et al., 2023) and DLinear (Zeng et al., 2023), experience a notable performance drop on the PeMS datasets, which are characterized by strong spatiotemporal dependencies. In contrast, although **IndexNet** also adopts a CI modeling approach, it achieves consistently strong performance across all four PeMS benchmarks. This improvement can be attributed to the incorporation of spatiotemporal prior information via timestamp and variable index embeddings. For instance, on the PeMS04 dataset, IndexNet reduces MAE and RMSE by 19.2% and 23.1% compared to PatchTST, and by 18.4% and 21.2% compared to DLinear. Similarly, on PeMS07, IndexNet achieves a 27.5% reduction in MAE compared to PatchTST, and a 15.1%

| Models | Metric | IndexNet | SCINet | Crossformer | PatchTST | TimesNet | MICN | DLinear | iTransformer | Autoformer | Informer |
|---|---|---|---|---|---|---|---|---|---|---|---|
| | MAE | **15.18** | 15.64 | 15.97 | 18.95 | 16.41 | 15.71 | 19.70 | 16.72 | 18.08 | 19.19 |
| PeMS03 | MAPE | **15.16** | 15.89 | 15.74 | 17.29 | 15.17 | 15.67 | 18.35 | 15.81 | 18.75 | 19.58 |
| | RMSE | **24.16** | 24.55 | 25.56 | 30.15 | 26.72 | 25.20 | 32.35 | 27.81 | 27.82 | 32.70 |
| | MAE | **19.57** | 20.35 | 20.38 | 24.86 | 21.63 | 21.62 | 24.62 | 21.81 | 25.00 | 22.05 |
| PeMS04 | MAPE | **12.15** | 12.84 | 12.84 | 16.65 | 13.15 | 13.53 | 16.12 | 14.85 | 16.70 | 14.88 |
| | RMSE | **31.14** | 32.31 | 32.41 | 40.46 | 34.90 | 34.39 | 39.51 | 33.91 | 38.02 | 36.20 |
| | MAE | **20.66** | 22.28 | 22.54 | 27.87 | 25.12 | 22.79 | 28.65 | 23.01 | 26.92 | 27.26 |
| PeMS07 | MAPE | **8.55** | 9.38 | 9.41 | 12.69 | 10.60 | 9.57 | 12.15 | 10.02 | 11.83 | 11.63 |
| | RMSE | **33.42** | 35.40 | 35.49 | 42.56 | 40.71 | 35.61 | 45.02 | 35.56 | 40.60 | 45.81 |
| | MAE | **15.17** | 17.38 | 17.56 | 20.35 | 19.01 | 17.76 | 20.26 | 17.94 | 20.47 | 20.96 |
| PeMS08 | MAPE | **9.64** | 10.76 | 10.92 | 13.15 | 11.83 | 10.80 | 12.09 | 10.93 | 12.27 | 13.20 |
| | RMSE | **24.17** | 27.34 | 27.21 | 31.04 | 30.65 | 27.26 | 32.38 | 27.88 | 31.52 | 30.61 |

Table 8: Short-term forecasting results in the PeMS datasets.

reduction in RMSE compared to DLinear. Furthermore, IndexNet even surpasses recent channel-dependent (CD) methods such as iTransformer (Liu et al., 2024c) and Crossformer (Zhang & Yan, 2023), demonstrating its effectiveness in capturing complex spatiotemporal correlations despite its CI architecture. These results highlight the robustness and generalization ability of IndexNet in real-world traffic forecasting scenarios.

## B.8 ABLATION STUDY

| TE | CE | Length | ETTm1 MSE | ETTm1 MAE | Weather MSE | Weather MAE | Solar MSE | Solar MAE | Electricity MSE | Electricity MAE | Traffic MSE | Traffic MAE |
|---|---|---|---|---|---|---|---|---|---|---|---|---|
| | | 96 | 0.328 | 0.365 | 0.175 | 0.215 | 0.229 | 0.266 | 0.164 | 0.250 | 0.434 | 0.276 |
| | | 192 | 0.363 | 0.380 | 0.222 | 0.256 | 0.263 | 0.289 | 0.173 | 0.259 | 0.446 | 0.281 |
| × | × | 336 | 0.395 | 0.404 | 0.278 | 0.296 | 0.287 | 0.303 | 0.190 | 0.276 | 0.462 | 0.288 |
| | | 720 | 0.459 | 0.442 | 0.354 | 0.347 | 0.286 | 0.301 | 0.229 | 0.309 | 0.493 | 0.306 |
| | | *Avg.* | 0.386 | 0.398 | 0.257 | 0.278 | 0.266 | 0.290 | 0.189 | 0.274 | 0.459 | 0.288 |
| | | 96 | 0.326 | 0.365 | 0.162 | 0.207 | 0.224 | 0.262 | 0.150 | 0.240 | 0.432 | 0.274 |
| | | 192 | 0.359 | 0.379 | 0.207 | 0.248 | 0.256 | 0.285 | 0.163 | 0.252 | 0.444 | 0.280 |
| × | ✓ | 336 | 0.388 | 0.401 | 0.263 | 0.289 | 0.280 | 0.299 | 0.180 | 0.269 | 0.460 | 0.286 |
| | | 720 | 0.448 | 0.438 | 0.346 | 0.344 | 0.282 | 0.300 | 0.222 | 0.306 | 0.494 | 0.305 |
| | | *Avg.* | 0.380 | 0.396 | 0.245 | 0.272 | 0.260 | 0.287 | 0.179 | 0.267 | 0.458 | 0.286 |
| | | 96 | 0.311 | 0.353 | 0.168 | 0.207 | 0.202 | 0.244 | 0.152 | 0.241 | 0.392 | 0.258 |
| | | 192 | 0.365 | 0.383 | 0.216 | 0.250 | 0.233 | 0.267 | 0.163 | 0.251 | 0.397 | 0.266 |
| ✓ | × | 336 | 0.394 | 0.406 | 0.273 | 0.293 | 0.246 | 0.279 | 0.180 | 0.269 | 0.418 | 0.273 |
| | | 720 | 0.453 | 0.442 | 0.351 | 0.345 | 0.260 | 0.283 | 0.219 | 0.302 | 0.453 | 0.290 |
| | | *Avg.* | 0.381 | 0.396 | 0.252 | 0.274 | 0.238 | 0.269 | 0.178 | 0.266 | 0.415 | 0.272 |
| | | 96 | 0.312 | 0.352 | 0.155 | 0.198 | 0.192 | 0.230 | 0.137 | 0.230 | 0.384 | 0.253 |
| | | 192 | 0.355 | 0.379 | 0.202 | 0.243 | 0.219 | 0.255 | 0.154 | 0.245 | 0.391 | 0.260 |
| ✓ | ✓ | 336 | 0.386 | 0.402 | 0.260 | 0.287 | 0.239 | 0.271 | 0.171 | 0.262 | 0.411 | 0.270 |
| | | 720 | 0.445 | 0.437 | 0.343 | 0.341 | 0.241 | 0.269 | 0.212 | 0.299 | 0.459 | 0.286 |
| | | *Avg.* | **0.374** | **0.392** | **0.240** | **0.267** | **0.223** | **0.256** | **0.169** | **0.259** | **0.411** | **0.256** |

Table 9: Ablation on the effect of removing TE and CE sub-modules. ✓ indicates the use of the sub-module, while × means removing the sub-module. Results are averaged from all prediction lengths.

| TE | Length | ETTm1 | | Weather | | Solar | | Electricity | | Traffic | |
|---|---|---|---|---|---|---|---|---|---|---|---|
| | | MSE | MAE | MSE | MAE | MSE | MAE | MSE | MAE | MSE | MAE |
| Zeros Init | 96 | 0.312 | 0.352 | 0.155 | 0.198 | 0.192 | 0.230 | 0.137 | 0.230 | 0.384 | 0.253 |
| | 192 | 0.355 | 0.379 | 0.202 | 0.243 | 0.219 | 0.255 | 0.154 | 0.245 | 0.391 | 0.260 |
| | 336 | 0.386 | 0.402 | 0.260 | 0.287 | 0.239 | 0.271 | 0.171 | 0.262 | 0.411 | 0.270 |
| | 720 | 0.445 | 0.437 | 0.343 | 0.341 | 0.241 | 0.269 | 0.212 | 0.299 | 0.459 | 0.286 |
| | *Avg.* | **0.374** | **0.392** | **0.240** | **0.267** | **0.223** | **0.256** | **0.169** | **0.259** | **0.411** | **0.256** |
| Random Init | 96 | 0.321 | 0.360 | 0.159 | 0.201 | 0.202 | 0.246 | 0.142 | 0.233 | 0.693 | 0.460 |
| | 192 | 0.367 | 0.384 | 0.210 | 0.249 | 0.230 | 0.269 | 0.158 | 0.249 | 0.650 | 0.453 |
| | 336 | 0.387 | 0.402 | 0.271 | 0.295 | 0.258 | 0.288 | 0.174 | 0.264 | 0.680 | 0.465 |
| | 720 | 0.447 | 0.443 | 0.351 | 0.347 | 0.261 | 0.289 | 0.216 | 0.303 | 0.751 | 0.501 |
| | *Avg.* | 0.380 | 0.397 | 0.246 | 0.273 | 0.237 | 0.272 | 0.172 | 0.262 | 0.694 | 0.470 |
| Latent Addition | 96 | 0.315 | 0.356 | 0.155 | 0.199 | 0.198 | 0.245 | 0.139 | 0.230 | 0.405 | 0.261 |
| | 192 | 0.358 | 0.381 | 0.203 | 0.244 | 0.228 | 0.268 | 0.156 | 0.247 | 0.409 | 0.265 |
| | 336 | 0.391 | 0.408 | 0.262 | 0.289 | 0.258 | 0.288 | 0.174 | 0.266 | 0.422 | 0.276 |
| | 720 | 0.448 | 0.439 | 0.347 | 0.346 | 0.261 | 0.290 | 0.210 | 0.299 | 0.462 | 0.291 |
| | *Avg.* | 0.378 | 0.396 | 0.241 | 0.269 | 0.236 | 0.272 | 0.170 | 0.260 | 0.424 | 0.273 |
| Channel Projection | 96 | 0.348 | 0.380 | 0.174 | 0.224 | 0.254 | 0.293 | 0.169 | 0.269 | 0.602 | 0.351 |
| | 192 | 0.382 | 0.391 | 0.222 | 0.268 | 0.301 | 0.326 | 0.181 | 0.275 | 0.612 | 0.362 |
| | 336 | 0.414 | 0.412 | 0.282 | 0.309 | 0.322 | 0.357 | 0.198 | 0.303 | 0.632 | 0.360 |
| | 720 | 0.481 | 0.455 | 0.369 | 0.364 | 0.341 | 0.355 | 0.224 | 0.318 | 0.654 | 0.373 |
| | *Avg.* | 0.406 | 0.410 | 0.262 | 0.291 | 0.304 | 0.333 | 0.193 | 0.291 | 0.625 | 0.362 |

Table 10: We conduct ablation studies to investigate the impact of timestamp embedding (TE) modeling strategies. **Zeros Init** refers to initializing the TE vectors with zeros, while **Rand Init** denotes random initialization. **Latent Addition** represents an early-stage timestamp modeling approach, where timestamp embeddings are added to the latent projected input sequence. **Channel Projection** represents another early-stage timestamp modeling approach, where timestamp embeddings and the original multivariate series are projected along the channel dimension to fuse temporal information.

**Embedding Strategies.** To compare the effectiveness of different timestamp embedding strategies, we conduct ablation studies as shown in Tab. 10. We first compare two initialization methods for learnable timestamp embeddings: zero initialization and random initialization. The results indicate that zero initialization generally yields more stable and superior performance, while random initialization may introduce noise that leads to performance degradation, particularly on complex datasets like Traffic. Injecting timestamp embeddings into the latent space also causes a slight performance drop, possibly due to misalignment between the projected representations and temporal positions. Finally, adding timestamp information after channel projection leads to the most significant decline, suggesting that improper integration of temporal cues can hinder model performance, consistent with recent findings (Wang et al., 2024a).

