# OpenReview forum: "IndexNet: Timestamp and Variable-Aware Modeling for Time Series Forecasting"
_ICLR.cc/2026/Conference — ICLR 2026 Conference Withdrawn Submission_

### Official Review · Reviewer_zjSB · 2025-10-19

**Soundness:** 2
**Presentation:** 3
**Contribution:** 2
**Rating:** 4
**Confidence:** 4

**Summary:**

IndexNet proposes an MLP-based framework for multivariate time series forecasting, incorporating timestamp and variable index information through an Index Embedding module. The model demonstrates competitive performance on various datasets and offers interpretability through visualization of embeddings. However, the novelty and motivation of the approach require further justification, and the theoretical foundation could be strengthened.

**Strengths:**

**Index Embedding:** The use of separate timestamp and variable index embeddings offers a unique perspective on incorporating index-related information into MTSF models, potentially capturing temporal and variable-specific dynamics effectively.

**Competitive Performance:** IndexNet achieves competitive results compared to state-of-the-art models on diverse datasets, including both long-term and short-term forecasting tasks.

**Analysis:** The visualization of embeddings provides valuable insights into the learned temporal patterns and variable relationships.

**Weaknesses:**

**Limited Innovation:** The proposed approach builds upon existing MLP-based architectures, with the Index Embedding module being a relatively straightforward extension.

**Weak Motivation:** The paper lacks a strong motivation for the proposed approach. A more detailed discussion on the limitations of existing methods and the specific benefits of incorporating index information would strengthen the paper’s rationale.

**Theoretical Foundation:** The paper lacks a thorough theoretical foundation for the proposed method. Exploring the theoretical underpinnings of the Index Embedding module, such as its connection to existing time series analysis techniques, would enhance the paper’s depth.

**Limited Effectiveness:** While the model demonstrates competitive performance, the improvements over existing methods are marginal.

**Questions:**

Same as Weaknesses.

---

### Official Review · Reviewer_wzzB · 2025-10-27

**Soundness:** 4
**Presentation:** 3
**Contribution:** 3
**Rating:** 6
**Confidence:** 5

**Summary:**

This paper presents **IndexNet**, an MLP-based model for multivariate time-series forecasting that explicitly encodes **Timestamp** and **Channel Identity** via **Timestamp Embedding (TE)** and **Channel Embedding (CE)**. The MLP backbone captures short-term periodic-pattern capture with low computational cost; TE models long-term seasonalities, while CE distinguishes channel identities. Experiments show that IndexNet delivers superior performance with low computational cost and nice interpretability.

**Strengths:**

1. The paper is well organized: the method is clearly presented and the experiments are reasonably complete.

2. The design is simple and efficient; visualizations of TE/CE convincingly demonstrate the model's interpretability.

3. The method explicitly models timestamps and channel identities—the key factors often overlooked yet intuitively crucial for forecasting.

4. Strong results on an MLP backbone, with consistent gains when integrated into CNN and Transformer backbones, showing nice generalization ability.

**Weaknesses:**

1. Evaluation is largely confined to week/day/hour/minute cycles on relatively small datasets spanning only a few years, making the effectiveness on month/year seasonality insufficiently validated.

2. Real-world periodicities are not always cleanly cyclic (e.g., week definitions vary; year sequences may have no natural endpoint). The paper would benefit from discussing or testing non-cyclic/irregular encodings.

3. TE/CE yield striking gains on the MLP backbone but relative small improvements on CNN/Transformer backbones. Deeper analysis (e.g., interaction with attention/convs, capacity-matched baselines) would clarify the bottlenecks and strengthen the contribution.

**Questions:**

1. As paper note (L246–L258), current evaluated datasets are small and short, making year/month seasonality hard to assess. Can you validate on larger, longer-span datasets?

2. How to encode years in practice? Real data rarely form clean year-to-year cycles. What is your strategy for non-cyclic or drifting yearly patterns?

3. TE/CE yield strong gains on the MLP backbone but relative smaller improvements on CNN/Transformer baselines. What explains this gap (e.g., redundancy with attention/convs, capacity limits, training dynamics)?

4. Clarify differences from related work—especially methods in Figure 1(b). What is difference in IndexNet's TE/CE design compared with prior works(e.g. GLAFF, Autoformer), and how does it alter the modeling pipeline?

5. In the NIPS 2024 workshop [1], some researchers pointed out that current methods sometimes use the "drop-last" trick [2] to improve performance. Therefore, It is recommended that you clarify whether the "drop - last" operation was used in your paper in the implementation details section of your paper for transparency.

[1] Fundamental limitations of foundational forecasting models: The need for multimodality and rigorous evaluation

[2] TFB: Towards Comprehensive and Fair Benchmarking of Time Series Forecasting Methods

---

### Official Review · Reviewer_gtK3 · 2025-10-30

**Soundness:** 3
**Presentation:** 3
**Contribution:** 2
**Rating:** 4
**Confidence:** 3

**Summary:**

**Problem.** In multivariate time-series forecasting (MTSF), two major paradigms exist: Channel-Independent (CI) and Channel-Dependent (CD). The authors argue that existing methods either ignore the explicit timestamp semantics or overlook variable identity (channel index), limiting interpretability and robustness.

**Method.** The paper proposes **IndexNet**, a lightweight MLP-based forecasting model enhanced with an **Index Embedding (IE)** module, including:

- **Timestamp Embedding (TE):**
  Learns embeddings from discrete calendar fields such as minute/hour/day/week/month. These embeddings are indexed by timestamp and summed, then concatenated with the projected input representations. (Fig. 2, §3.3)

- **Channel Embedding (CE):**
  Assigns a learnable identity vector to each variable, concatenated with time-aware features to distinguish channels explicitly. (§3.4)

- **Backbone:**
  A residual MLP stack (two Linear + ReLU layers), followed by a linear prediction head for horizon *T*. (§3.5)

**Results.** Evaluated on 12 datasets (Tab. 1, Tab. 7), IndexNet reportedly achieves comparable or better SOTA performance on average, especially with long lookback windows (L = 336) while reducing GFLOPs and inference latency (Tab. 3). Ablations (Tab. 2‒3, Tab. 9‒10) show both TE and CE contribute positively. PCA visualizations (Fig. 3‒4) claim to capture periodicity and variable similarity.

**Claimed contributions**:
A general, interpretable, plug-and-play module (TE + CE) that works efficiently within CI/MLP frameworks for MTSF.

**Strengths:**

- Simple & efficient: MLP + embedding concatenation yields strong latency/complexity profiles
- Broad validation across 12 datasets and multiple horizons
- Plug-and-play potential demonstrated on different models
- Visual results improve interpretability for timestamp embedding

**Weaknesses:**

### (a) Inconsistent logic and notation
- §3.3 states that each time embedding has dimension **L**, but the formulas later use **T_dim**, and concatenation yields **d_model + T_dim** (p.5).
  This conflicts with Fig. 2 and raises uncertainty about how TE aligns with the temporal dimension.
- Subscripts of \( z, z_t, z_{t,c} \) are semantically inconsistent across §3.3‒3.5 (time steps vs. variable-level features), creating ambiguity for implementation.

### (b) Complexity
- The training cost (like duration, complexity) compared to previous work was not calculated.

### (c) Ambiguity in plug-and-play evaluation
- In Tab. 4 (§4.5), IE / TE / CE are stacked alongside other “plugins” (GALF / VH), but the label semantics are confusing, making it difficult to quantify the **stand-alone** contribution of IE.

### (d) Prior art not properly acknowledged
The claim “timestamps and variable identity are ignored by existing methods” is overstated. Many works already incorporate:

- **Time embeddings** in MTS models (e.g. Informer/TimesNet)
- **Plug-and-play timestamp modules** (e.g., GLAFF)
- **Learnable semantic time vectors** (Time2Vec, D2Vformer)

**Verdict:** Method is reasonable, but the paper needs clearer formulation, more rigorous and fair comparisons, and more accurate positioning in related work.

**Questions:**

1. **Novelty claim inflation & insufficient related-work positioning**
   Must clarify differences vs. GLAFF, Time2Vec, D2Vformer.

2. **Dimension & alignment not rigorously defined**
   TE shape and its sliding-window handling require precise tensor diagrams or pseudocode.

3. **Generalization to new or reordered variables**
   CE depends on fixed indices—robustness to unseen channels not discussed.

4. **Used of featrures**
   Why were these calendar features chosen from the paper, and why weren't other features like "quarters," "holidays," and "Month of Year" included? It is unclear whether **all** baselines receive the same exogenous features; otherwise results are asymmetric.

**Details Of Ethics Concerns:**

1. **Novelty overstated relative to existing literature**
   The paper frames the incorporation of timestamp semantics and variable identity as a neglected problem. However, these ideas have been widely explored:
   - Timestamp embeddings in Transformer-based MTS models (e.g., Informer, Autoformer, TimesNet)
   - Learnable timestamp representations (Time2Vec, D2Vformer)
   - Plug-and-play timestamp modules (GLAFF)
   The contribution appears incremental rather than conceptually new.

2. **Inconsistency and ambiguity in tensor formulations**
   §3.3 introduces contradictory claims regarding TE dimension (**L** vs. **T_dim**) and the concatenated shape (**d_model + T_dim**).
   This creates uncertainty in how TE interacts with:
   - Temporal axis alignment
   - Sliding-window inference
   - Positional encodings
   Implementation could easily diverge from author intentions.

3. **Fairness issues in comparative studies**
   - Traffic dataset uses extended external features (year/month), but it is unclear if **all** baselines were given the same inputs.
   - Table 4 does not isolate the independent contribution of IE (TE/CE), confounding plugin combinations with other modules.

4. **Generalization and robustness concerns**
   - CE strictly binds embeddings to observed channel indices.
     → What happens with **unseen variables**, reordered channels, or dynamic sensor sets?
   - Timestamp construction via modulo could lead to misalignment across splits or test-time calendar shift.

5. **Lack of direct comparisons against most relevant baselines**
   Given the claim of plug-and-play design, a fair validation should compare **within identical backbones** against:
   - Time2Vec / D2Vformer
   - GLAFF (timestamp plugin)
   Without such ablations, the empirical advantage of IE remains unclear.

---

### Official Review · Reviewer_owj6 · 2025-11-01

**Soundness:** 3
**Presentation:** 3
**Contribution:** 1
**Rating:** 2
**Confidence:** 4

**Summary:**

The paper proposes **IndexNet**, a multivariate time series forecasting model designed to effectively incorporate **timestamp** and **variable index** information using a lightweight MLP-based framework. It introduces an **Index Embedding (IE)** module, consisting of **Timestamp Embedding (TE)** and **Channel Embedding (CE)**, to improve prediction accuracy and model interpretability by leveraging the temporal and variable-specific patterns in the data. The model achieves competitive forecasting performance across multiple real-world datasets, demonstrating robustness and efficiency, especially in capturing periodic and heterogeneous dynamics.

**Strengths:**

1. **Well-written**: The paper is clearly written, with a well-structured presentation of the methodology and results. The descriptions of the model and its components, such as the **Index Embedding (IE)** module and the **Timestamp Embedding (TE)** and **Channel Embedding (CE)** submodules, are clear and detailed, making it easy to follow the approach and understand its contributions.

2. **Significant Experimental Results**: The experiments show that **IndexNet** outperforms several state-of-the-art models across a variety of real-world datasets, demonstrating its effectiveness in improving forecasting accuracy and reliability. The results are consistent and robust across different domains, validating the model’s ability to capture both temporal and variable-specific dynamics effectively.

**Weaknesses:**

Here is the revised and translated version of your review:

---

**Weakness**:
My primary concern lies in the similarity between this paper and STID \[1]. Although the paper cites STID, the methodology section appears to be almost identical to STID. The main difference between IndexNet and STID seems to be the addition of a few timestamp features. If that is indeed the case, I believe the innovation in this paper is nearly nonexistent and may raise potential ethical concerns.

\[1] Spatial-Temporal Identity: A Simple yet Effective Baseline for Multivariate Time Series Forecasting. CIKM 2022.

---

This version maintains the essence of your review while improving clarity and flow in English.

**Questions:**

See weakness.

---

### Note · Authors · 2025-11-27

I have read and agree with the venue's withdrawal policy on behalf of myself and my co-authors.